# Association of Proteins Modulating Immune Response and Insulin Clearance during Gestation with Antenatal Complications in Patients with Gestational or Type 2 Diabetes Mellitus

**DOI:** 10.3390/cells9041032

**Published:** 2020-04-21

**Authors:** Arthur T. Kopylov, Anna L. Kaysheva, Olga Papysheva, Iveta Gribova, Galina Kotaysch, Lubov Kharitonova, Tatiana Mayatskaya, Anna Krasheninnikova, Sergey G. Morozov

**Affiliations:** 1Institute of General Pathology and Pathophysiology, Department of Pathology, 125315 Moscow, Russia; a.a.krasheninnikova@yandex.ru (A.K.); sergey_moroz@list.ru (S.G.M.); 2Institute of Biomedical Chemistry, Department of Proteomic Researches, 119121 Moscow, Russia; kaysheva1@gmail.com; 3Sergey S. Yudin 7th State Clinical Hospital, Perinatal Center, 115446 Moscow, Russia; gkb29@mail.ru; 4Nikolay E. Bauman 29th State Clinical Hospital, 110020 Moscow, Russia; iveta.gribova@yandex.ru (I.G.); suslatin1@mail.ru (G.K.); 5“Biopharm-Test” Limited Liability Company, 121170 Moscow, Russia; 6Nikolay I. Pirogov Medical University, 117997 Moscow, Russia; luba2k@mail.ru (L.K.); taska@mail.ru (T.M.)

**Keywords:** diabetic fetopathy, gestational diabetes mellitus, type 2 diabetes mellitus, insulin resistance, immunoglobulins, carnosine

## Abstract

Background: The purpose of the study is to establish and quantitatively assess protein markers and their combination in association with insulin uptake that may be have value for early prospective recognition of diabetic fetopathy (DF) as a complication in patients with diabetes mellitus during gestation. Methods: Proteomic surveying and accurate quantitative measurement of selected proteins from plasma samples collected from the patients with gestational diabetes mellitus (GDM) and type 2 diabetes mellitus (T2DM) who gave birth of either healthy or affected by maternal diabetes newborns was performed using mass spectrometry. Results: We determined and quantitatively measured several proteins, including CRP, CEACAM1, CNDP1 and Ig-family that were significantly differed in patients that gave birth of newborns with signs of DF. We found that patients with newborns associated with DF are characterized by significantly decreased CEACAM1 (113.18 ± 16.23 ng/mL and 81.09 ± 10.54 ng/mL in GDM and T2DM, *p* < 0.005) in contrast to control group (515.6 ± 72.14 ng/mL, *p* < 0.005). On the contrary, the concentration of CNDP1 was increased in DF-associated groups and attained 49.3 ± 5.18 ng/mL and 37.7 ± 3.34 ng/mL (*p* < 0.005) in GDM and T2DM groups, respectively. Among other proteins, dramatically decreased concentration of IgG4 and IgA2 subclasses of immunoglobulins were noticed. Conclusion: The combination of the measured markers may assist (AUC = 0.893 (CI 95%, 0.785–0.980) in establishing the clinical finding of the developing DF especially in patients with GDM who are at the highest risk of chronic insulin resistance.

## 1. Introduction

Diabetic fetopathy (DF) is a neonatal disease that is developed in newborns and characterized by systemic lesions, metabolic and endocrine dysfunctions and caused by condition of type 1 diabetes mellitus, type 2 diabetes mellitus (T2DM) or gestational diabetes mellitus (GDM) suffered by pregnant women. Diabetic fetopathy is the cause of premature birth, chronic hypoxia, respiratory depression, asphyxia of the fetus at birth, accompanied by various metabolic disorders, and can also be the cause of newborns death [1,2,3]. This is one of the most common forms of obstetric complications and perinatal losses evoked from hyperglycemia manifested in a disturbance of glucose tolerance and occurred or was first recognized during pregnancy [4]. It is also possible that the disturbance of carbohydrate metabolism could precede pregnancy but has not been previously detected [5,6]. During pregnancy, hyperglycemia is induced by a gradually increased insulin resistance due to both placental hormones and the mother’s hormones (prolactin, estrogen and cortisol). The growing concentration of these hormones is compensated by a decrease in the clearance of endogenously produced insulin [2,5].

According to the WHO data for 2016, up to 25% of pregnant women are at risk of hyperglycemia during pregnancy [7]. An overwhelming majority (from 75% to 90%) of elevated blood glucose levels during pregnancy emerges from the already progressing GDM [8,9]. Statistics of the IDF for 2017 demonstrate up to 204 million women aged between 20 to 79 years live with different types of diabetes, while up to 21.3 million newborns have symptoms of hyperglycemia, which is 85.1% of cases was stipulated by GDM [8,9]. According to the statistical data in Russia for 2017, hyperglycemia complication during pregnancy contributes 36.5% of DF cases [10]. In turn, 87% of pregnant women with GDM are defined by the rising level of C-peptide and 93% by increased insulin resistance index, in contrast to uncomplicated gestation, where these indicators are 4% and 6%, respectively (*p* < 0.05) [9]. Since GDM during pregnancy is a direct indicator of the high risk of DF [1,2,3,11], further signs observed by ultrasound examination can be an indication for management by insulin therapy.

Despite rigorously scrutinized pathogenesis of DF, its prevention and diagnosis remain an acute point upon a present time. The imperfection of methods for the accurate diagnosis of hyperglycemia does not allow the prevention of DF on time [12]. The key reason for the poor sensitivity is the lack of reliable clinical markers that would permit recognition the risk of DF, especially, in the early gestational age. Clinical methods for treatment and prevention of DF are limited to continual monitoring of maternal blood glucose level, diet intervention or insulin therapy, but the effectiveness is debatable and depends largely on the term for detection of DF [13].

The most crucial determinants of the impact on fetal growth and development caused by a diabetic condition during pregnancy can be predicted in the first trimester. Recently, much has been discovered about how maternal lipids metabolism can affect fetal adipose tissue development. The congenital abnormalities of fetal growth correlate with maternal BMI and plasma triacylglycerol and involve complex regulation through PPAR (peroxisome proliferator-activated receptors) receptors [14,15].

Monitoring of maternal HbA1c can also be a predictor of fetal macrosomia which is one of the signs of diabetic fetopathy indicated during an ultrasound examination. It has been demonstrated that using only this indicator without combination with other markers, it is possible to predict GDM before 20 weeks, preeclampsia and even the risk of perinatal death [16]. Although the HbA1c is widely used and assessed by the WHO indicator, it is usually measured in the combination of insulin C-peptide and glucose levels [17]. Therefore, its predictive and diagnostic value as a sole indicator on the surface seems unlikely.

Magnetic resonance imaging (MRI) traits for assessment and precise monitoring of fetal growth is some kind specific compare to ultrasound examination, but unfortunately, this diagnostic strategy did not demonstrate more sensitive results to predict macrosomia [18].

The primary place for DF diagnosis is occupied by ultrasound examination during pregnancy [12,19]. However, the signs of DF cannot be reliably detected in the first trimester, despite recommendations to conduct such studies between 10–14 weeks of gestational age [19]. The DF can be asymptomatic even after ultrasound examination and biochemical tests up to 30 weeks of gestation, whereas from the 31st week there may be a surge in the size of fetus, accompanied by glycemic indicators at the lower limits [8,11,12,19]. Over the second and third trimesters, it is possible to recognize the fetal development syndrome as a consequence of the progressing hyperinsulinemia. In this case, the main indicators are expressed in the increased size of the fetus by nearly 2 weeks ahead of the actual period, polyhydramnios and disproportion in the size [12,19]. Thus, none of the existing methods provide sufficient and reliable signs of emerging DF during pregnancy.

The affected newborns are morphologically and functionally immature and do require a staged treatment. The main phenotypic signs of DF in newborns are overweight, facial edematous, pronounced shoulder girdle and cardiomyopathy [1,3]. The metabolic finding of the pathology is an attitude in the underestimated level of blood glucose, elevated hemoglobin and difficulty in breathing. Soon after birth, the newborn may experience neurological disorders like insomnia, decreased muscle tone and abrupt activity changes [1,8].

This study aimed to identify coherent markers for the assessment of DF complications and to recognize the molecular events underlying the reasons for DF. Observed proteins correlate well with metabolic pathways affecting the lipid, immune system and carbohydrate balance, and allows to determine pitfalls in violations of these pathways. In contrast to traditionally employed instrumental methods for diagnosis of DF, which are generally sensitive in late gestational age, the obtained results may be useful for a deeper understanding of the molecular pathophysiology and for predicting diabetic fetopathy.

## 2. Materials and Methods

### 2.1. Ethical Considerations

Design of the study and use of human material was approved by local research ethics committee of the N.E. Bauman 29th Hospital (Moscow) and conducted in accordance with the WMA Declaration of Helsinki on Ethical Principles for Medical Research Involving Human Subjects (over the study completion the actual version was revision Fortaleza, 2013).

### 2.2. Population

The total population was comprised of 179 subjects who participated in the study during their routine clinical screening in N.E Baumann State Clinical Hospital between 2017 and 2018 years over gestation. The BMI (body mass index) was measured between 16–19 weeks of gestation.

Patients of GDM positive group (n = 80) were selected based on the results of OGTT and were subdivided into those who delivered healthy newborns (group G01; n = 43, age 27.2 ± 4.8 years, BMI 26.48 ± 5.41 kg/m^2^) and who delivered newborns with signs of diabetic fetopathy (group G02; n = 37, age 26.7 ± 5.4 years; BMI 26.77 ± 4.99 kg/m^2^).

Patients with T2DM (n = 63) were selected based on the clinical records and history of pre-existing and manifesting type 2 diabetes mellitus, and were subdivided in the same way as for GDM positive patients: a subgroup of patients delivered healthy newborns (G03; n = 34, age 27.7 ± 5.9 years, BMI 30.99 ± 3.29 kg/m^2^) and a subgroup delivered newborns with signs of diabetic fetopathy (G04; n = 29, age 25.2 ± 5.1 years, BMI 28.86 ± 6.20 kg/m^2^).

The control group (group G05; n = 36, age 26.6 ± 5.2 years, BMI 21.68 ± 4.25 kg/m^2^) was enrolled from women with an uncomplicated pregnancy course and gave birth of healthy newborns with no signs of fetopathy during ultrasound examination and postpartum.

#### 2.2.1. Inclusion Criteria for Patients with GDM or T2DM

The GDM observation was carried out between 23–28 gestation weeks using a 75 g OGTT (oral glucose tolerance test) according to recommendations of IADPSG (revision 2010) [20] and adopted by Russian Association of Obstetrician and Gynecologist (revision 2012) [21] as follows: the fasting glucose level should not exceed 5.8 mmol/L and patients whose OGTT in one hour was below 9.8 mmol/L and had no previous history of any type of diabetes mellitus in clinical records were considered as the normal glycemic study group. Patients whose OGTT in one hour exceeded 9.8 mmol/L were passed to complete the next 2 h of OGTT for a final establishing of GDM and were considered as GDM-positive if OGTT showed more than 8.5 mmol/L. (Table 1, groups G01 and G02). The ratio of insulin-treated GDM patients and dietary-intervention strategy was 27% and 78%, correspondingly. Patients with GDM were treated during gestation by either of the following medication: Insuman® Rapid GT or NovoRapid® FlexPen® at a dose of 7–14 IU per day (totally) of the short-acting insulin over gestation. Four patients with the previous history of GDM (*n* = 3 of 6 in G01 and *n* = 1 of 3 in G02, Table 1) were established as GDM-positive at 6–8 weeks of gestational age (I trimester) based on the fasting glucose level varied from 6.8 mmol/L to 8.5 mmol/L and prandial glucose level varied between 9.5 mmol/L to 11.4 mmol/L. The strategy of insulin treatment for such patients started from 7–14 IU per day in the first trimester and continued in the second and third trimesters at a dose of 25–30 IU per day. Insulin was delivered (bolus) three times per day (at 8 a.m., 1 p.m. and 6 p.m.) before meals for all patients under consideration.

Patients with T2DM were selected according to their clinical history of type 2 diabetes mellitus and associated disorders (Table 1, groups G03 and G04). In total, the duration of diabetes mellitus was 7.4 ± 3.8 years from the first clinical record of T2DM manifestation. All the patients of G03 and G04 groups (Table 1) with T2DM had a previous history of insulin therapy with an average dose of insulin 7.0 ± 2.2 units per day according to their clinical records and interview survey. A history of cardiovascular disease or any other chronic disease, including inflammatory and autoimmune diseases were excluded at the initial stage of observation. Patients with pre-eclampsia condition manifested during gestation were excluded from the study cohort throughout the assay.

#### 2.2.2. Diagnostic Criteria for Diabetic Fetopathy

Fetal biometry by sonography examination was performed using an Acuson 128 XP4 ultrasound machine (Siemens Inc., Munich, Germany), equipped with a 3.5-MHz probe. Signs of fetopathy were established between 22–35 gestational weeks age according the following criteria: discreet fetal growth until 30 weeks, enlargement of the abdomen along with glycemic values at the lower limit, excessive macrosomia until 35 weeks; primary established polyhydramnios if other reasons except GDM were not established; hepatosplenomegaly. Patients were stratified according the clinical manifestation of diabetic fetopathy resulted by sonography examination and postpartum (Table 1). Evaluation and reporting the status of the newborn at birth (including Apgar-1 and Apgar-5 scores) is given in the Appendix A.

### 2.3. Reagents

Urea (99%) and formic acid (98%+, pure) were obtained from Acros Organics (Geel, Belgium). Trifluoroacetic acid (99%, Reagent Plus®), triethylammonium bicarbonate (1 M solution), 4-vinylpyridine (95%), sodium deoxycholic acid (>97% titration) were from Sigma (St. Louis, MO, USA). Acetonitrile (HPLC grade, filtered for 0.2 µm) was purchased from Fisher Chemical (Loughborough, UK). Acetic acid (EMSURE®, glacial, anhydrous for analysis) was from Merck (Darmstadt, Germany). TCEP (Tris(2-carboxyethyl)phosphine hydrochloride) was purchased from Pierce™ (Thermo Fisher, Rockford, IL, USA). Trypsin (sequencing grade modified) was supplied by Promega (Madison, WI, USA). Water (TOC<3 ppb) was obtained from Milli-Q Integral 3 purification system, Millipore S.A.S (France).

### 2.4. Blood Samples Collection and Handling

Peripheral venous blood samples (4–6 mL of maternal blood, labeled as M-series samples) were collected from the patients into EDTA-2K^+^ Vacutainer plasma tubes (BD, USA) following an overnight fasting (approximately 12 h) in the morning between 8 a.m. to 10 a.m. The blood samples were processed according to the manufacturer’s instructions and centrifuged at 10 °C and 2500 *g* for 10 min. A volume of 0.5–1.0 mL of the collected plasma samples was retrieved specifically for proteomic study, filtered through 0.22-µm cellulose–acetate filters and stored at −80 °C until use. The total protein concentration in plasma samples was determined by the BCA (Smith assay) Protein Assay Kit ab102536 (Abcam, Cambridge, MA, USA). Plasma samples (100 µg of total proteins fraction equal to 1.7–3.0 µL of plasma depending on the certain sample) were diluted to 20 µL of 5-M urea, 7.5-mM TCEP, 100-mM TEABC and 1% sodium deoxycholic acid. Samples were incubated for 30 min at 40 °C. Alkylation was performed by adding 2 µL of 2% solution of 4-vinylpyridine in 30% isopropanol and incubation for 20 min at ambient temperature in darkness. Samples were diluted to 200 µL finally by 100-mM TEABC, and trypsin (1 µg per a sample) was added. Digestion was carried out at 37 °C for 3 h following additional of trypsin in amount of 500 ng and incubation for the next 3 h at 37 °C. The reaction was stopped by adding 10 µL of 10% formic acid and centrifuged at 10,000 *g* at 10 °C to sediment deoxycholic acid. Samples were dried under vacuum at 30 °C for 90 min, then pellets were reconstituted in 100 µL of 0.5% formic acid.

### 2.5. Liquid Chromatography and Surveying Mass Spectrometry

Peptides were separated using liquid chromatography on an Ultimate 3000 RSLC Nano (Thermo Scientific, Waltham, MA, USA). Samples were loaded onto an Acclaim Pepmap® (5 mm × 0.3 mm, 300-Å pore size, 5-µm particle size) column for 4 min at a flow rate of 15-µL/min in a mobile phase C (water with 3.5% acetonitrile supplied with 0.1% formic acid and 0.05% acetic acid). Peptides were washed out from the enrichment column and separated onto analytical column Acclaim Pepmap® (75 µm × 150 mm, 1.8-µm particle size, 60 Å pore size) at a flow rate of 0.30 µL/min in a gradient of mobile phase A (water) and mobile phase B (90% acetonitrile and 10% methanol) both supplied with 0.1% formic acid and 0.03% acetic acid. Post-run column equilibrating in the initial gradient condition was performed for 12 min.

Mass spectrometry analysis was performed on a high-resolution Orbitrap Fusion (Thermo Scientific, Waltham, MA, USA) mass spectrometer. Precursor ions were surveyed at a resolution of R = 60 K in a range of 425–1250 *m/z*. Ions were isolated using quadrupole with +1.0 Th and +0.25 Th offset isolation window and a maximum integration time of 15 ms. Ions with charge states of z = 2+…7+ were triggered for tandem scanning. Fragment ions were obtained at 27% normalized HCD activation energy and detected in an ultra-high field orbital mass analyzer at a resolution of R = 15 K. Ions were accumulated for a maximum integration time of 47 ms. The complete single duty cycle time was 4 sec.

### 2.6. Accurate Quantitative Mass Spectrometry Analysis

The quantitative analysis was designed using UPS-2 (Proteomics Dynamic Range Standard Set) as described in [22,23]. The UPS-2 standard contains 48 defined proteins covering a dynamic range from 0.5 fmoles to 50,000 fmoles (total amount is proteins is 10.6 µg; the proteins and their characteristics of proteins are listed in Appendix A). Because the UPS-2 contains proteins which are generally residents of human plasma, it was prepared as the standards set fortified with *Escherichia coli* (strain K12) used as a matrix to avoid interference with endogenous human plasma proteins. The matrix and UPS-2 were prepared separately as described in Section 2.4 for the studied samples. The final concentration of the UPS-2 standard proteins prepared for mass spectrometry analysis was ranged between 0.05 fmoles/µL to 5000 fmoles/µL and the estimated concentration of *Escherichia coli* (strain K12) proteins used as a matrix was 1 µg/µL. The hydrolyzed UPS-2 sample was spiked into the matrix at a ratio of 1:10 so to make the final estimated range proteins concentration from 0.005 fmoles/µL to 500 fmoles/µL. Two µL of the prepared sample was run in six technical replicated in the same gradient as indicated in the Section 2.4. The obtained data were analyzed against the proprietary custom FASTA-file available on http://www.sigma.com/ups to identify the proteins annotated in the UPS-2 in the same way as described in the Section 2.7. Quantitative analysis was performed based on the selection of proteins identified in each of the six replicated with the standard deviation of intensities not exceeded 20% (and no more than 10% at 500 fmoles/µL calibration point). In total, 27 proteins were plotted to construct the calibration curve covering the concentration range from 0.005 fmoles/µL to 500 fmoles/µL against the normalized total intensities of the corresponding proteins in the UPS-2 (Appendix A). The obtained calibration curve was fitted to the linear regression with r^2^ = 0.9128 (see Appendix A). Proteins of interest, obtained after identification in the analyzed samples, were plotted on the fitted calibration curve and the protein concentration were estimated in the same fashion as was performed for the UPS-2 proteins using normalized total intensities of specifically unique peptides.

### 2.7. LC-MS Data Analysis and Proteins Identification

Data files obtained after data-dependent LC-MS/MS analysis was converted in peak list format and used for proteins identification. Peak lists obtained from spectra were identified using OMSSA version 2.1.9. The search was conducted using Search GUI version 2.3.17 [24]. Protein identification was conducted against a concatenated target/decoy database of human proteins (Uniprot, release June 2018). The decoy sequences were created by reversing the target sequences in Search GUI. The identification settings were as follows: trypsin as a specific protease, with a maximum of 1 missed cleavage, tolerance of ±5 ppm as MS1 level and tolerance of ±0.0025 Da as MS2 level tolerances; variable modifications: oxidation of M (+15.994915 *u*), deamination of N (+0.984016 *u*), deamination of Q (+0.984016 *u*). Peptides and proteins were inferred from the spectrum identification results using Peptide Shaker version 1.16.15 [25]. Peptide Spectrum Matches (PSMs), peptides and proteins were validated at a 1.0% false discovery rate (FDR) estimated using the decoy hit distribution [26].

### 2.8. Statistical Data Analysis and Functional Clustering Analysis.

Analysis of categorical traits was performed using the Kruskal–Wallis test and differences were considered as significant for *p* < 0.05. Due to the small size of the studied groups, significance in the measured proteins concentration between groups was evaluated by Fisher’s exact test with significance level cut-off of *p* < 0.05. Bias-correction of intra- and intergroup comparisons were evaluated using Spearman’s and Kendal’s rank correlation test to exclude outliers and to reveal the possible impact of medication strategy (insulin therapy and dietary intervention) for GDM positive patients. Mann–Whitney test used for the evaluation significance of anthropometric and biochemical parameters between studied groups at *p* < 0.05.

Meaningful proteins were quantified using the UPS-2 standard and were subjected to the area under the receiver operating curve (auROC) analysis in the R (version 3.2.0). The auROC was calculated separately for each protein to estimate their specificity and sensitivity and as an integrative value for the combination of proteins produced the most significant impact.

To elucidate the changes in the circulating proteome of patients, we provided a comparative analysis of groups stratified by the criterion of pregnancy complication (gestational or type 2 diabetes mellitus) or by the criterion of pregnancy outcome (healthy newborn or with signs of diabetic fetopathy). Proteins shared between the analyzed groups we extracted, ranged according to their representation (based on the NSAF (Normalized Spectral Abundance Factor) values obtained after proteins identification) and submitted for Gene Ontology analysis with the functional annotation tool of the PANTHER Overrepresentation Test (annotation release 20181113) with the Fisher’s exact test against the list of proteins in the control group (used as a reference list) to unveil underlying biologic processes and functions associated with significantly changed proteins [27]. Bonferroni correction for multiple testing was selected for the estimation of the p-value and FDR. The resulting GO terms of each analyzed subset with significance *p* < 0.001 were refined through REVIGO opensource [28] using similarity coefficient 0.7 (medium) which is capable to summarize significant terms in way of their meaningful and remove the redundant ones. Refined GO terms were associated with the listed proteins and further analyzed in support of the Gorilla open-source [29] to visualize the hierarchical clusterization of the meaningful biologic processes in terms of GO with the p-value threshold at least *p* < 0.001. Human molecular pathways were extracted from the KEGG [30] and the Reactome (version 65 Released 20180612) databases [31].

## 3. Results

The main clinical and anthropometric characteristics of patients are given in Table 1. There were no significant differences between groups in age, gestational age at delivery and in results of OGTT examination for GDM positive groups. Significance was determined in BMI between examined groups (G01–G04) and the control group of patients with an uncomplicated pregnancy and varied between *p* = 0.003 to *p* = 0.02 (Table 1). Patients with GDM (G01 and G02) and T2DM (G03 and G04) did not differ significantly in BMI (*p* = 0.75). Samples were collected at the gestational age of 23–28 weeks and no significant differences of OGTT in GDM patients were inspected. The results of OGTT in fasting glucose were slightly lower in the GDM group compared to patients of the T2DM group and both significantly higher than in the control group G05 (Table 1). The strategy of GDM management (insulin treatment or dietary intervention) did not display significant differences between such patients (Kendall’s *tau* rank correlation returned coefficient t^2^ = 0.881, the Pearson’s correlation returned correlation of r^2^ = 0.851) therefore these patients were not stratified according to the strategy of GDM treatment and were not considered separately.

The proteomic survey revealed 321 proteins shared between all studied groups. Further consideration was focused on a piece of the proteome that was specifically recognized for semi-quantitative alterations between groups under consideration and comprised of 75 different proteins. Most the identified proteins (apolipoproteins family, elements of complement cascades and hemostasis, transporting proteins) are generally mentioned to a variety of even unrelated disorders. Moreover, the difference between almost 87% proteins of this shared proteome was insignificant (*p*-values varied from 0.27 to 0.83 at cut-off *p* < 0.01). Yet small portion of proteins revealed a significance (*p* < 0.01) but had an insufficient frequency (from 0.02 to 0.72) while we considered proteins must were featured by a frequency of 1.0 due to the small size of the studied groups (Table 1) in our research. Although we support the role of most these proteins in the pathogenesis of diabetes mellitus and their possible impact on fetus growth and development, we incline to consider these proteins as the secondary response effectors with low (or insufficient) specificity and selectivity. In this way, we limited only a small fraction of proteins that met the criteria of (1) sufficient frequency, (2) rare mentioning in regard of other pathologies, (3) relation to the disturbed insulin metabolism and accompanied immune reaction and (4) possibility to connect proteins of interest into proposed molecular mechanism reflecting the process of insulin resistance and associated diabetic fetopathy. At the same time, it was demonstrated that GDM groups (G01 and G02, Appendix A and Appendix A) are featured by the largest number of specifically altered proteins: there were 56 proteins attributed for G01 (healthy newborns) whereas 37 proteins were specifically revealed for G02 (newborns with signs of DF). The number of proteins that uniquely ascribed to T2DM groups was rather low (10 and 34 for G03 and G04 groups, correspondingly; Appendix A and Appendix A).

Proteomes between groups with T2DM and GDM are fundamentally different in their biologic processes by the represented group-specific proteins (Figure 1). The vast majority of proteins in GDM groups (G01 and G02) were classified as modulators of metabolic macromolecular complexes with a positive effect (GO:0010604, *p* = 8.38 × 10^−4^), while in the groups with T2DM (G03 and G04) most proteins referred to the transport and cell-functional proteins (GO:0006810, *p* = 8.24 × 10^−4^) (Appendix A). The more comprehensive separation between pathology groups displayed in their cellular localization (Table 2). It was found that T2DM groups (G03 and G04) displayed proteins that were defined as intracellular localization (GO: 0044424, *p* = 8.17 × 10^−4^), while groups with GDM (G01 and G02) exhaustively piled up toward extracellular localization (GO: 0070062, *p* = 3.73 × 10^−6^).

Among proteins commonly observed between studied groups, only a few could display confident quantitative differences and may determine a possible impact on the development of diabetic fetopathy. These proteins were quantitatively measured using the UPS-2 calibration and normalization approach (Figure 2 and Appendix A). Apparently, the selected promising proteins should contribute to glucose uptake and concomitant insulin clearance. In this way, we chose CNDP1 and CEACAM1 proteins among most significantly alternated between groups and participated in insulin clearance and resistance mechanisms (Table 3).

In the presented research we detected a significantly lower concentration of CEACAM1 in the G02 group (113.18 ± 16.23 ng/mL, *p* = 0.003) and less extensive in the G04 group (81.09 ± 10.54 ng/mL, *p* < 0.001) compare to the G05 group with uncomplicated gestation (515.6 ± 72.14 ng/mL). Both groups (G02 and G04) fitted to different types of diabetes mellitus but gave birth of newborns with sings of DF meaningfully differed with groups where diabetic patients characterized by the normal course of fetus development (*p* = 0.012 in G02 vs G01 and G03 and *p* = 0.008 in G04 vs G01 and G03).

Patients with T2DM and GDM are characterized by significantly increased tissue glycolysis following acidification by excessively produced lactate. The CNDP1 was found as a meaningful protein but its concentration between groups of patients with T2DM was characterized by less difference (*p* = 0.076 between G03 and G04 groups) compare to patients with GDM (*p* = 0.041 between G01 and G02 groups). The measured concentrations of CNDP1 in groups with DF-signed newborns were 49.3 ± 5.18 ng/mL in G02 (*p* = 0.019 compare the control) and 37.7 ± 3.34 ng/mL in G04 group (*p* = 0.033 compare the control). Although it was significantly higher compared to the control group G05 where CNDP1 reached up to 17.1 ± 4.31 ng/mL (*p* = 0.021, G05 vs G02 and G04) less prominent differences as indicated in comparison with groups G01 and G03 where diabetic patients delivered healthy newborns (*p* = 0.041 in G02 vs G01 and G03 and *p* = 0.061 in G04 vs G01 and G03; Table 3).

The risk of insulin resistance and T2DM is associated with the development of chronic inflammation and extensive immune response. We suggested to focus attention on immunoglobulins and their ratios and CRP was selected as a general marker of the inflammatory condition. Expectedly, CRP returned a prominent difference in its level between groups stratified by signs of diabetic fetopathy. The measured concentration of CRP was of 5.29 ± 1.82 μg/mL (*p* = 0.038 for G02 vs G01 and G03; *p* < 0.01 for G02 vs control group G05) and 4.21 ± 1.37 μg/mL (*p* = 0.024 for G04 vs G01 and G03; *p* = 0.011 for G04 vs G05) in groups of GDM and T2DM that gave birth of newborns with clinical signs of DF, whereas groups with healthy newborns demonstrated CRP comparable with the levels in control group G05 and made 2.04 ± 1.32 μg/mL (*p* = 0.54 for G01 vs G05) and 2.84 ± 0.67 μg/mL (*p* = 0.47 for G03 vs G05; Table 3).

A similar phenomenon can be inspected for Ig-proteins (Table 4). The IgG4 was the only that was reduced and reached a minimum in groups with DF (1.85 mg/mL in G02 (*p* = 0.031) and 1.81 mg/mL in G04 (*p* = 0.024) vs 2.64 mg/mL in the control group G05). At the same time, only IgG1 displayed a meaningful increase in G01 (*p* < 0.01) and G03 (*p* = 0.013) groups with healthy newborns compared to the control group G05. There were almost no variations in concentrations of IgG2 and IgG3 throughout the study groups compare to baseline (p-value varied from 0.371–0.737 compare to the G05 group; Table 4).

On the contrary, depending on the GDM or T2DM condition, the level of IgM increased up to 1.67–1.93 mg/mL. The most prominent distinction was detected for group G02 compare the control group G05 (*p* = 0.039) and compare the groups of diabetic patients with healthy newborns (*p* = 0.026 for G02 vs G01 and G03; Table 4) where its concentration raised significantly. Similar behavior IgM displayed in the G04 group where the concentration was expressively higher compare to the control group G05 (*p* = 0.025) and slightly but still meaningfully decreased compare to the groups of diabetic patients with healthy newborns (*p* = 0.038 for G04 vs G01 and G03; Table 4).

Since IgA proteins were measured deferentially as for IgG proteins, we observed that the concentration of IgA2 is significantly increased (up to 2-fold) in all studied groups. The most prominent elevation of IgA2 was featured in groups with signs of DF and reached 1.98 mg/mL (G02, *p* = 0.013) and 1.55 mg/mL (G04, *p* = 0.033) compare to the control group G05 (Table 4). The significance of IgA1 was also meaningful for groups of patients who characterized by newborns with signs of DF (*p* = 0.029 for G02 vs G01 and G03 and G05; *p* = 0.037 for G04 vs G01 and G03 and G05).

The discriminating ability of the measured proteins (CEACAM1, CNDP1, CRP, IgG4 and IgA2) that improves is demonstrated as the integral receiver operating curve (ROC) with AUC = 0.893 (95% CI, 0.785–0.980) for all five measured markers (Figure 3). The most substantial input in overall selectivity and specificity was conducted by CEACAM1 and CNDP1, making these two proteins as the most promising markers, while CRP holds a position of auxiliary marker providing delicate correction in the final prognostic panel. The total specificity of 0.923 and selectivity of 0.891 endorses a high potency of the selected markers for the prediction of diabetic fetopathy condition in patients with T2DM and GDM during gestation.

## 4. Discussion

The observed distribution of proteins suggests that despite different clinical manifestations in the studied groups, only a small part of the proteome reflects the molecular mechanisms involved in the development of DF. Apparently, it indicates the absence of fundamental rearrangements between proteomes architecture of patients with T2DM and GDM and in associated metabolic processes (Appendix A).

Network analysis of protein–protein interactions (PPI) did not demonstrate significant results that could reflect the maturation of signaling pathways within any of the studied groups. The most significant result at utmost confidence (*p* = 0.081) was reached in the case of the G02 group. The analysis of GO clustering also returned unsatisfied results on the grouping of the proteins according to their molecular functions. However, in regard to the distribution of proteins according to their involvement in biologic processes (Figure 1) and attributed cellular localization we obtained engrossing results (Table 2).

Distinct distribution of proteins by subcellular localization may determine the specificity of cellular metabolic processes that underline the difference in the origination of T2DM or GDM (Table 2). In particular, exosomes and extracellular localization of proteins (for example, TUBB; EZR, CD14, LTF) in GDM groups exhibit extensive processes matrix architecture remodeling as a consequence of prominent production of reactive oxygen species (QSOX1, SOD2, SEPP1) and pro-inflammatory condition.

An overwhelming majority of routing indicators (Table 2, Appendix A) are already known as exact association with diabetes. Previously, proteome-scaled profiling of serum in pregnant women with GDM revealed several promising biomarkers including AFM, VTN, APCS, SERPINC1, etc. However, after validation by SRM mass spectrometry, only VTN has been confirmed as most contributing to the maternal risk factor of GDM development [32]. Other proteomic investigations delivered results that could be associated with GDM onset through the disturbances of the coagulation process, inflammatory condition, immune response, complement activation, oxidative stress, etc. [33]. Some proteins (including CDKN2A, CDKN2B, HHEX, ENPP1, PPARG) have been repeatedly identified as potential determinants of GDM and its dire consequences for the fetal growth [33,34]. Selection of only four the most promising proteins (APOE, F9, FGA and IGFBP5) validated by ELISA, entailed to the conclusion about impairment of lipids metabolism as the key process determining the GDM and its consequences [33]. Yet other proteomic research supported by TMT-labeling observed both up- and down-regulated proteins, whereas only two of them (FLT1 and PABPC4) were verified by immunoblotting. Further functional analysis suggested that GDM is caused by oxidative stress, cell migration, angiogenic disorder, etc. [35].

Turning to the panoply of the evidence-based results, it seems that interplay of lipid transport, oxidative stress, activation of the complement system and associated impairment of hemostasis make the main contribution in the development of chronic inflammation that, consequently, influences on the remodeling of matrix architecture. These enrolled disturbances may affect morphologic development of the placenta and malfunction of glucose and triglycerides placental transport which produces noticeable sensation in course of the most critical period of fetal development (from 22–24 weeks of gestation).

In this research, quantitatively measured proteins were accounted as elements of molecular events associated with GDM and fetus development. We paid special attention to carnosine dipeptidase-1 or CNDP1 (EC 3.4.13.20), shared between groups with DF (G02 and G04). This protein is a member of the metabolic transformation of alanine, β-alanine and histidine and is a primary link in the recycling of carnosine (βAla-His), anserine and some other dipeptides. Carnosine is a buffering metabolite and can bind hydrogen from lactate during glycolysis finalization and to stabilize the tissue environment from oxidative damage (Figure 4) [36]. The substrate for CNDP1 is mainly found in skeletal muscle tissue while the most abundant gene expression was demonstrated in the brain [37].

Association of CNDP1 with diabetes mellitus has been widely demonstrated in many papers [38,39,40]. In proteomic experiments based on the iTRAQ measurements, this protein has been discovered as a promising marker involved in T2DM patients and demonstrated lowering (adjusted *p* = 3.4 × 10^−5^) after a very low-calorie diet [41]. Still, it is debatable which organ is the first target of insulin resistance, it seems likely that skeletal muscle may play a crucial role in this process (since it mediates over 70% of all insulin-mediated glucose disposal). Although it has been shown that carnosine is indeed reduced in diabetic model rats [42] and diabetic liver [43], these data are not always significant [44] and contradict with other findings made for the skeletal muscle [45].

We observed that the measured concentration of CNDP1 was upregulated in all groups characterized by DF (Table 3). However, patients with T2DM were characterized by less difference in concentration of CNDP1 compare to GDM (Table 3), assumingly, due to activation of compensatory mechanisms engaging transport proteins (AFM, FBLN1 and APOM) for controlling the blood glucose levels [46,47,48].

The relationship between metabolic pathways of glucose and carnosine is not evidently established [58], therefore it is fair to assume that elevated levels of CNDP1 are a secondary response in consequence of the progressing GDM or T2DM. Given impaired lipids and glucose metabolism in such patients, the oxidative glucose utilization is diminished towards the increased tissue glycolysis [59]. The following acidification by excessively produced lactate and interconversion of lactate and pyruvate is greatly enhanced in patients with T2DM and GDM [60]. Catabolism of carnosine is being occurred with a higher rate, which can explain the increased concentration of CNDP1 in the blood of such patients.

Although there is no strong evidence about the association of upregulated CNDP1 and the increased lactate level, significant association of insulin resistance caused by increased glycolysis in muscle and decreased aerobic capacity has been demonstrated [55]. Hence, the increased plasma lactate may improve the responsiveness to insulin and promote its secretion. Meanwhile, lactate concentrations are chronically increased in diabetic patients with obesity when hyperlactatemia in obese individuals was found to be preceding the diabetes onset process [56].

Upon closer examination, it should be noted that in our study the concentration of CNDP1 is quantitatively higher for GDM patients who underwent dietary intervention therapy rather than those managed by insulin treatment but still both groups remained the equal probability of complicated fetus developing (Table 1 and Table 3). The enzyme CNDP1 has been mentioned concerning T2DM, where carnosine acted as a protective factor in diabetic nephropathy [61]. Hence, increasing the CNDP1 in patients with T2DM and GDM can bear the protective effect in response to insulin resistance.

The Carcinoembryonic antigen-related adherence cell-1 (CEACAM1) or CD66a, is a protein attracting extensive attention due to its multipotent ability in the regulation of insulin signaling and lipogenesis [62]. This protein belongs to the CEA family (carcinoembryonic antigens), which are diverged into a group of adhesion protein molecules and a group of glycoproteins specific for the gestation period. The CEACAM1 is enormously polymorphic and currently known 11 isoforms are yielded as products of alternative splicing. The main function of the protein is focused on homophilic cell adhesion along the Ca^2+^-independent pathway [63]. It can also act as a co-inhibitory receptor in the immune response and such property is accomplished through the mechanism of phosphorylation or via the PTPN6-dependent pathway, leading to a reverse reaction (dephosphorylation) downstream to the effectors’ cascade [64].

Recently, it has been demonstrated that complete mutation in CEACAM1 gene (*Ccl*^-/−^) brings to impartments of insulin clearance causing hyperinsulinemia and limited insulin resistance [65]. Follow-up insulin stimulation, the CEACAM1 undergoes active phosphorylation at Ser508 by insulin receptors (INSR) [50]. The interaction of CEACAM1 occurs through the binding to the SH2-domain of SCH-1 protein (Figure 4). This ultimately contributes to active insulin clearance through the mechanism of receptor-mediated endocytosis [50,51]. In turn, the internalization of insulin results in the interaction of CEACAM1 with FASN (fatty acid synthase) stipulating negative regulation of fatty acid synthesis by diminishing FASN activity [53] By this way, CEACAM1 mediates the co-inhibitory effect of insulin signaling and promotes the down-regulatory outcome of insulin on gluconeogenesis and lipogenesis.

Among 11 known CEACAM1 isoforms [66], only isoform-1 possesses the ability to interact with FASN through the insulin-dependent mechanism and exactly that isoform was detected in the samples of studied groups [50,53]. Turning to the obtained results of quantitative measurements, we assumed that the suppression of CEACAM1 has appeared as a consequence of GDM and T2DM manifestation (Table 3 and Figure 4). Both CNDP1 and CEACAM1 can be tracked along the chain of events that enhance association with DF especially in patients with GDM (Table 3). During gestation the level of CEACAM1 is being gradually increased, nevertheless, the GDM complication is typically manifested in increasing insulin resistance which may be caused by a significant deficiency of CEACAM1 (Table 3 and Figure 4). The following-up impairment of glucose uptake leads to the shift toward non-oxidative glucose utilization. Insofar glycolysis rate is being increased substantially, the concentration of lactate and its interconversion dramatically grow [67,68]. This entails the increased synthesis of carnosine of which metabolism is regulated by CNDP1 and carnosine synthase. Hence, both CEACAM1 and CNDP1 can be considered as potential hallmarks for assessment of GDM progression and their ratio may indicate the possible progression of dire consequences such as DF.

Among other proteins, the CRP or C-reactive protein, should also be mentioned. The CRP is widely used as a reliable marker for monitoring of the inflammatory response of various etiologies as well as for monitoring the treatment carried out by steroids and NSAIDs [69,70]. This protein varied in a wide range throughout the studied groups. Since CRP fluctuates at many different conditions, it is hard to be accounted for a proper indicator of the certain antenatal pathology under consideration.

However, there are data on the association of an elevated level of CRP with rising insulin resistance [71]. Additionally, data extracted from the STRING and KEGG databases demonstrate that both, CRP and insulin, participate in acute inflammatory reaction and the regulation of the immune effector response (GO:0002673 and GO:0002697). According to the WHO data on the 2010 year [72], the risk of insulin resistance and T2DM is associated with the development of chronic inflammation, regardless of its etiology. Some authors reported a direct relationship between an increased level of CRP and progressing insulin resistance in patients with diabetes mellitus (CRP concentration 3.84 ± 1.45 μg/mL) [73]. Other authors asserted about the strict dependence of the developing insulin resistance in T2DM patients on the increased level of CRP. Moreover, it was established that the level of CRP more than 2.53 μg/mL in women (median age of 39 years, *p* < 0.001) raises the risk of insulin resistance 2.18-fold higher [72].

The primary reason for growing insulin resistance is a boosted activation of cytokines during acute and chronic inflammation [73,74]. In our study, the measured concentration of CRP was more than 2-fold higher in groups of patients that gave birth of newborns with signs of DF (G02 and G04) compare to groups with healthy newborns (Table 3). The obtained condition-casted distribution of the data suggests that in some cases an elevated level of CRP may be associated with GDM or T2DM, but significantly growing concentration, in turn, may be correlated with the complicated fetus development [75].

Variation of immunoglobulins concentration is typically observed in patients with diabetes mellitus, including GDM patients [76,77]. However, data presented in plenty of studies are somewhat contradictory. Case study of patients with type 1 diabetes mellitus showed a significant increase in the level of IgA (by +82.7%, *p* < 0.001) and IgG (+ 35.2%, *p* < 0.001), but a decrease in the level of IgM (by −6.7%, *p* < 0.001) and the described phenomenon is poorly explainable concerning to diabetes mellitus [76,78].

The obvious connection between the level changes of immunoglobulins and diabetes mellitus is not traced yet. It is known that the gestation is accompanied by such typical clinical findings as an increase in body weight and blood volume (approximately 10% and 20%, respectively), accompanied by alterations in the immune system, predominantly leading to mild suppression of the maternal immunity. Thus, it is expected that the level of Ig-proteins should be reduced in patients during gestation.

Notwithstanding, the level of total IgG was a little increased for groups with GDM and T2DM compared to the control group (Table 4). If we differentiate the measured IgG immunoglobulins by subclasses, oddly, the IgG4 was the only that has been reduced in all studying groups compared to the control group. Its minimum values were achieved in the groups with DF and made 2.12 mg/mL in G02 (*p* = 0.031) and 2.08 mg/mL in G04 (*p* = 0.024) vs 2.64 mg/mL in the control group G05. At the same time, a significant increase was observed only for the IgG1, while there were almost no variations for IgG2 and IgG3 subclasses (Table 4). On the contrary, the level of IgM was increased depending on the GDM or T2DM condition.

Other research group reported about various combinations of Ig-deficiency in patients with diabetes mellitus: patients with diabetes who have a history of infectious diseases were characterized by a reduced level of IgG, among then 25% of patients with a reduced level of IgA and almost 75% have a decreased level of IgM [79,80]. Since all these patients recently experienced various infectious diseases, it can be suggested that the combined effect of diabetes mellitus and immune response stimulated by infections affected the total profile of immunoglobulins. It should be considered as a pitfall because over gestation the immune response is reasonably weakened and fluxing [80,81].

In our study, the level of IgA was characterized by the highest level in groups with GDM and its moderate alterations were observed in groups with T2DM (Table 4). This comes in agreement with the data when the level of IgA was higher in healthy pregnancies and patients with GDM than in healthy non-pregnant patients [54]. However, patients associated with DF were featured by the highest abundance of IgA (Table 4). If we consider the concentrations of IgA1 and IgA2 separately, remarkably, the level of IgA2 is significantly upregulated (up to 2-fold) in all studied groups, whereas the most prominent elevation of IgA1 was observed only in groups with signs of DF (Table 4). On the other hand, an increased level of IgM and simultaneous depression in the production of immunoglobulins IgG is consistent with the suggestion that growing blood glucose levels can adversely affect the production of immunoglobulins.

It can be assumed that change in IgG and IgM levels is an indirect response of the immune system to diabetes (GDM or T2DM). Since IgG and IgA are the only classes of immunoglobulins exchanged through the placenta, a significant increase in IgG4 and IgA2 in patients with GDM can be considered as indicators of dramatically reduced transmission of the secondary immune response to a fetus due to progressing GDM and which may increase the possible risk of DF.

## 5. Conclusions

Only a small portion of the proteome can be confidently considered as potential markers of DF. The primary difference between T2DM and GDM groups was based on the subcellular localization of the group-specific proteins where GDM proteomic profile was characterized by numerous extracellular proteins, including exosome localization. Further consideration revealed a variety of markers unambiguously attributed to DF and these markers can be combined in line with both negative regulation of insulin signaling and shift the glucose uptake to non-oxidative utilization as hallmarks of insulin resistance.

Based on the evidence that anaerobic glycolysis retained an increasing rate in patients with T2DM and GDM, increased lactate producing and stressing of CNDP1, we assumed the involvement of carnosinase and its substrate in the proposed mechanism of insulin resistance (Figure 4). Disturbance of the insulin-mediated signaling is caused by substantially increased activity of liver CEACAM1 or CD66a, that mediates insulin clearance through INRS receptors. However, patients, who gave birth the newborns with signs of DF, were featured by a dramatic decrease of CEACAM1.

The concentration of CRP may progressively rise due to chronic inflammation and cytokines activation in both T2DM and GDM patients that may be extended to diabetic angiopathy, vessel damage and nephropathy caused by concomitant endocrine disturbances. We observed that patients with DF newborns were characterized by definitely increasing concentration of CRP which was at least 2-fold higher compare to uncomplicated pregnancy and groups with healthy newborns at delivery.

The level of immunoglobulins is complexed by gestation when the maternal immune system is partially suppressed, but we assume that decreased levels of exactly IgG4 and IgA2 can be reviewed as alarming indicators. The complimentary alteration of these two subclasses was strongly assigned to patients with newborns associated with DF.

The presented assembly of proteins encompasses typical markers that can be quantified in course of routine clinical analysis, however classical clinical chemistry devotes surprisingly little attention to the ratio of different immunoglobulins subclasses and auxiliary proteins responsible for insulin internalization and clearance. We found that proper combination (Figure 3) of widely known proteins can be associated with undesirable consequences affecting fetus development and, thus, may support the net benefit of traditional instrumental methods in the forecast of DF.

## 6. Limitations

Proteomic studies have a significant limitation stipulated by usually insufficient cohort size that limits translation of the obtained results and may weaken the proposed associations with the disease. Most of the proteomic researches are characterized by different sets of the identified proteins tending to take the role of meaningful biomarkers.

This cross-sectional study aimed to establish the relationship between diabetic fetopathy, and plasma proteins associated with regulation of insulin clearance, non-oxidative glycolysis and immune response. Until the lack of longitudinal study, the observed and measured proteomic markers cannot be considered as strong clinical markers for the prevention or the early diagnosis of DF, but rather they bear a supportive role in recognition of signs of DF and in stability of fetus development.

The obtained results may also be misrepresented by distinctions in selection criteria for the diagnosis of GDM between recommendations of the IADPSG, WHO and the National Consortium of the Russian Association of Obstetrician and Gynecologists. However, there are no fundamental distinctions between general and national recommendations and the rate of positive and negative results is expected to be equal.

Strategy in GDM management, i.e., therapy by insulin treatment or dietary intervention, may cause possible influence on the final results, although both approaches demonstrate almost equal efficiency in the treatment of GDM with unsuccessful rate outcome of almost 18% according to the WHO annual report on diabetes mellitus and its complication for the 2018 year. The possible influence of GDM management can be discovered across the significantly larger size of the study population and in combination with longitudinal monitoring of the important biochemical parameters, but at the current state of the research, we did not observe any meaningful difference between such patient treated by either insulin therapy or leading to dietary intervention.

## Figures and Tables

**Figure 1 cells-09-01032-f001:**
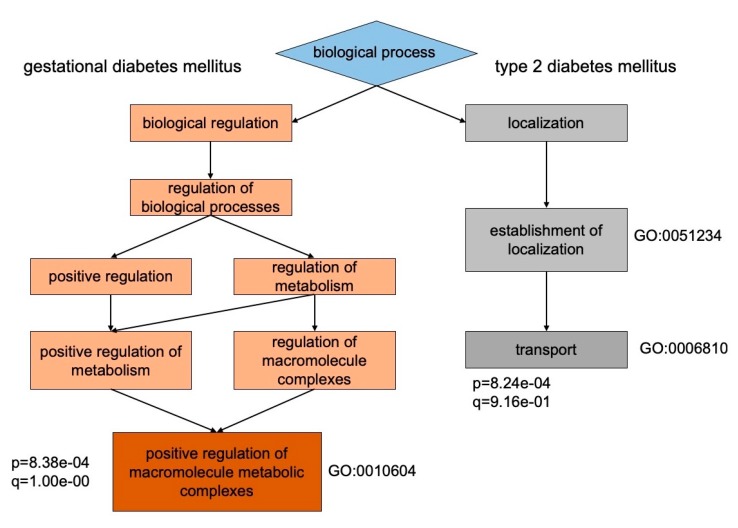
The hierarchical clustering of biologic processes for proteins determined specifically for groups with GDM and type 2 diabetes mellitus (T2DM). Patients with GDM are mostly characterized by proteins involved in positive regulation of macromolecule metabolic processes (GO:0010604, *p* = 8.38 × 10^−4^), whereas patients with T2DM are featured by transport proteins (GO:0006810, *p* = 8.24 × 10^−4^). This partially assigned to unequivocal mechanisms of chronic insulin resistance and glucose metabolism impairment between happened T2D and occurred GDM in the course of gestation.

**Figure 2 cells-09-01032-f002:**
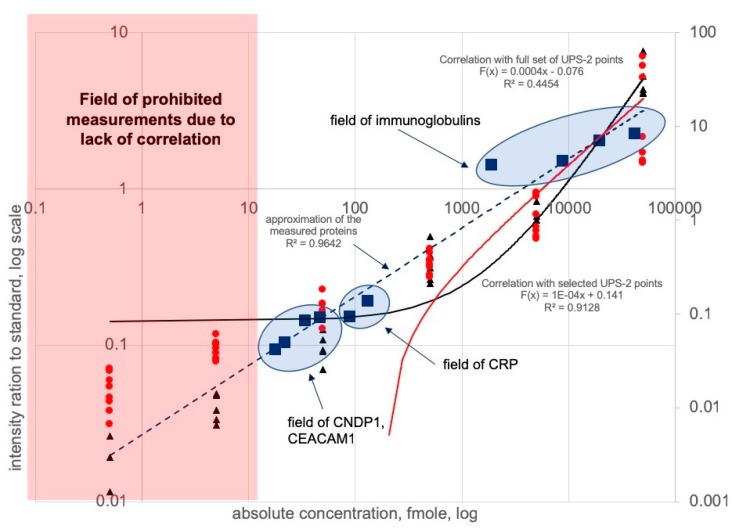
Calibration curve based on the measurements of the UPS-2 standards spiked into the non-human matrix. Red dots and red line indicate the complete set of calibration points plotted with correlation coefficient r^2^ = 0.44. Rational selection of the appropriate calibration points (black points and black line) enhanced the calibration and altered the correlation coefficient to r^2^ = 0.91. The prohibited margin for measurements was defined within 0.5–42 fmoles (red square) Proteins were measured within a range of 42–50,000 fmoles and plotted onto the selected correlation field indicated by blue circles. Approximation of the measured proteins CRP, CNDP1, CEACAM1 and immunoglobulins reached up to r^2^ = 0.96.

**Figure 3 cells-09-01032-f003:**
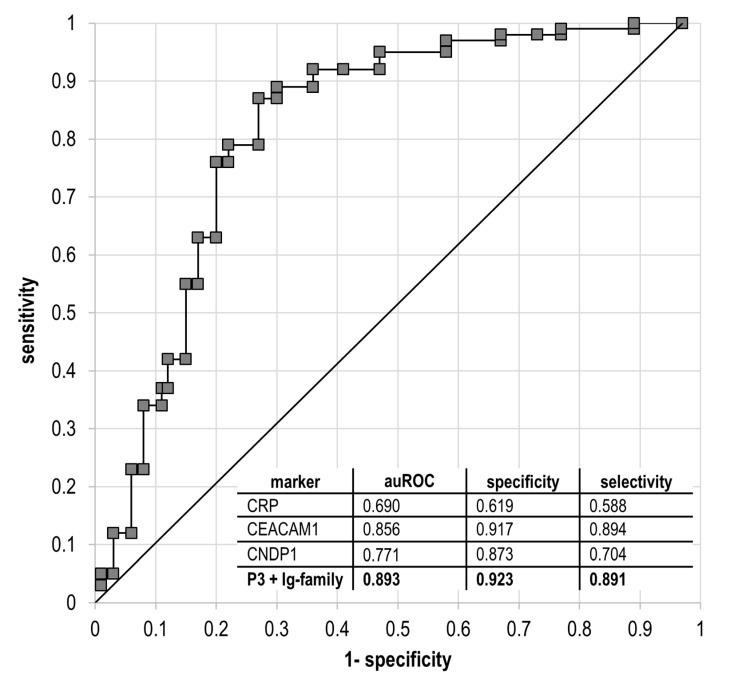
The integrative ROC curve for estimation of the diagnostic value of diabetic fetopathy in patients with T2DM or GDM complicated gestation using CEACAM1, CNDP1, CRP, IgG4 and IgA2 proteins. The integral AUC = 0.893 at 95% CI, 0.785–0.980. The least valuable protein is *CRP* caused by its highly sensitive fluctuations in response to impairments in many biologic processes (indicated in the figure insert). However, elimination of CRP from the proposed panel erodes the integral auROC to 0.813 (at 95% CI), hence CRP can be considered as an auxiliary marker among other more specific proteins in the panel. The proper combination of proteins in an appropriate quantitative range may endorse the ability of sensitive recognition of DF.

**Figure 4 cells-09-01032-f004:**
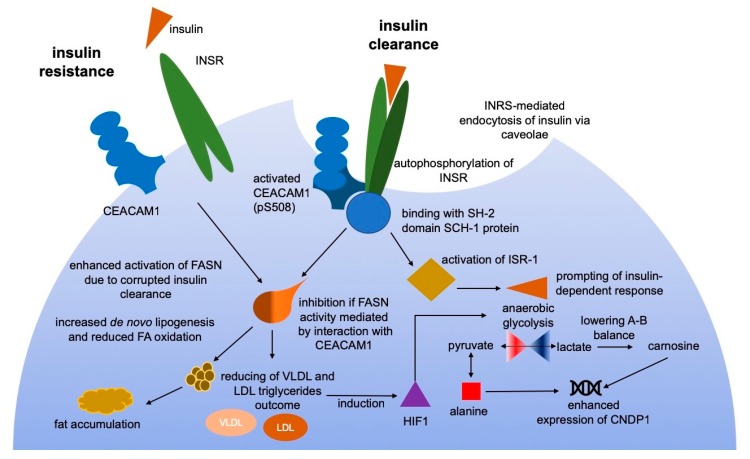
The proposed mechanism of insulin resistance in patients with T2DM and GDM. During uncomplicated pregnancy, CEACAM1 reaches the highest concentration in the third trimester [49]. Binding of insulin to INSR receptors promotes their autophosphorylation and downstream activation of CEACAM1 at position pSer508 and insulin-substrate receptor-1 (ISR-1) that enhances endocytosis of insulin [50,51]. Activated CEACAM1 may bind to the SH-2 domain-containing SCH-1 phosphatase leading to limitation of its activity and, thus, prolonging the phosphorylated state of CEACAM1. This entails to inhibition of fatty acid synthase (FASN) and increasing insulin-mediated response [52,53]. Corrupted CEACAM1-mediated insulin clearance follows to lowering fatty acids oxidation and to the enhancement of FASN activity [53,54]. Non-oxidative insulin-dependent utilization of glucose is typically enhanced in patients with T2DM and GDM. Growing interconversion of lactate and pyruvate is greatly increased and upregulates CNDP1 which is necessary for carnosine uptake [55,56]. The inhibited fatty acid oxidation concatenated by the hypoxia condition inducing hypoxia-inducible factor-1 (HIF-1) which triggers glycolysis in T2DM and GDM patients [57]. In turn, oxidative damage caused by lactate interconversion evokes the rising up-regulation of CRP as a protective response to defense against oxidative stress and subsequent chronic inflammation condition.

**Table 1 cells-09-01032-t001:** Summary of the main anthropometric data and measurements of glucose level for patients in study groups. The groups were aligned at BMI (body mass index) value which was measured between 16–19 weeks of gestational age. The GDM was established by 75-g OGTT (oral glucose tolerance test) conducted according the recommendations of IADPSG (revision 2010) [20] adopted by Russian Association of Obstetrician and Gynecologist (revision 2012) [21]. Diabetic fetopathy was diagnosed by ultrasound examination between 22–35 gestation weeks according to the inclusion criteria.

Group Number		G01	G02	G03	G04	G05	*p*-Value (Cut-Off *p* < 0.05)
Groups Description		GDM (27% Insulin-Treated; 78% Dietary Intervention)		T2DM		Uncomplicated Pregnancy	
Ultrasound Examination		Normal Course	Diabetic Fetopathy	Normal Course	Diabetic Fetopathy	Normal Course	
**Group size (n)**		43	37	34	29	36	0.174
**BMI ± SD, kg/m^2^ (between 16–19 weeks of gestation) ^#^**		26.48 ± 5.41 (*p* = 0.007) ^&^	26.77 ± 4.99 (*p* = 0.003) ^&^	30.99 ± 3.29 (*p* = 0.02) ^&^	28.86 ± 6.20 (*p* = 0.008) ^&^	21.68 ± 4.25	0.517
**Age ± SD, years**		27.2 ± 4.8	26.7 ± 5.4	27.7 ± 5.9	25.2 ± 5.1	26.6 ± 5.2	0.889
**OGTT (75 g), mean ± SD, mmol/L**	Fasting (>5.8 mmol/L) ^†^	6.1 ± 0.4 (*p* = 0.023) ^&^	6.0 ± 0.3 (*p* = 0.018) ^&^	9.7 ± 1.2 (*p* = 0.004) ^&^	9.9 ± 1.1 (*p* = 0.003) ^&^	3.8 ± 0.5	0.037
	1 h (>9.8 mmol/L) ^‡^	9.9 ± 1.1	10.6 ± 0.7	N/A	N/A	N/A	0.641
	2 h (>8.5 mmol/L) ^‡^	8.8 ± 0.4	9.2 ± 0.4	N/A	N/A	N/A	0.820
**HbA1c, mean ± SD, %**	I trimester	4.9 ± 0.3	5.6 ± 0.4	5.6 ± 0.5	5.5 ± 0.8	3.1 ± 1.5	0.917
	II trimester	6.7 ± 1.1	6.2 ± 2.0	8.2 ± 0.7	8.2 ± 2.5	3.5 ± 1.4	0.602
	III trimester	6.3 ± 0.7	5.9 ± 0.6	6.4 ± 1.0	6.8 ± 0.9	3.9 ± 0.9	0.873
**Caesar delivery, %**	emergency	2.33	8.11	0.00	0.00	5.56	N/A
	planned	27.91	29.73	29.41	34.48	36.11	N/A
**Gestational age at delivery, day; median (range)**		270 (259–278)	265 (258–270)	273 (268–289)	270 (269–284)	272 (265–288)	0.902
**Maternal weight gain, kg**		9.4 ± 4	11.7 ± 6	10.8 ± 5	12.1 ± 5	10.3 ± 6	0.673
**Fetal weight, kg**		3157 ± 254	4189 ± 212	3186 ± 198	4402 ± 278	3207 ± 112	0.044
**Family history of diabetes**		5	4	4	5	0	N/A
**Previous GDM history**		6	3	0	0	0	N/A

# difference in BMI within GDM groups and within T2DM groups is insignificant (*p* = 0.913 and *p* = 0.887, correspondingly); & significance of BMI and OGTT between the certain study group and control group (G05) evaluated by Mann–Whitney test; † difference in OGTT results at fasting glucose level is significant between GDM groups and T2DM groups (*p* = 0.0027); ‡ difference in OGTT results after 1 and 2 h of loading is insignificant between G01 and G02 groups (*p* = 0.993).

**Table 2 cells-09-01032-t002:** The identified proteins were clustered according to their cellular localization in terms of GO. Patients from the T2DM groups were assigned to intracellular localization while most proteins specified for GDM groups were characterized by extracellular and exosome localization.

Type of Diabetes Mellitus	GO Identifier	Annotation of Cellular Localization	*p*-Value	FDR *q*-Value	Enrichment Rate	Gene and Recommended Protein Names
T2DM	GO:0044424	intracellular part	8.2 × 10^−^^4^	1.7 × 10^−^^1^	1.14	ARL17B; CNGA2; CST3; IFI16; PPIA; GLTPD2; RIMS1; RPL13; GOLGA4; TEX26; GOLT1A; U2AF1L4; PACS2; CAPRIN2; COL6A3; EPB42; SEMA6D; CAT; HBM; DGKH; SARS; SPTB; SLC4A1; DENND1C; BLVRB; CDC40; ARHGAP18; ZNF883; ANK1; HSPA8
GDM	GO:0070062	extracellular and exosome	3.7 × 10^−^^6^	3.2 × 10^−^^4^	2.37	PDE8A; MASP2; SAA2; TUBB; EZR; QSOX1; IGF2R; LBP; CD14; SEPP1; RRAS2; SOD1; AGRN; TNXB; ABCB1; SERPINA5; LTF; TXN

**Table 3 cells-09-01032-t003:** Measured concentrations of CRP, CNDP1 and CEAEAM1 in groups with T2DM and GDM patients. Alternating concentrations probably does not permit to distinguish types of diabetes mellitus but allow proper recognition of groups with associated fetus development complication in form of diabetic fetopathy.

Acc. Number	Gene Name	Protein Name	G01, GDM	G02, GDM + DF	G03, T2DM	G04, T2DM + DF	G05, Control	*p*−Value †
P02741	CRP	C-reactive protein, mcg/mL	2.04 ± 1.32	5.29 ± 1.82	2.84 ± 0.67	4.21 ± 1.37	1.97 ± 0.71	3.12 × 10^−^^4^
P13688	CEACAM1	Carcinoembryonic antigen-related cell adhesion molecule 1, ng/mL	291.62 ± 34.55	113.18 ± 16.23	311.17 ± 42.16	81.09 ± 10.54	515.6 ± 72.14	5.33 × 10^−^^3^
Q96KN2	CNDP1	Beta-Ala-His dipeptidase, ng/mL	32.4 ± 5.23	49.3 ± 5.18	27.4 ± 2.63	37.7 ± 3.34	17.1 ± 4.31	2.17 × 10^−^^4^

† *p*-value was calculated by Kruskal–Wallis test at *p* < 0.05 significance; the indicated in Table *p*-value designates difference between all studied groups; statistical significance between the particular groups, if appropriate, is indicated in the text.

**Table 4 cells-09-01032-t004:** Concentrations of the detected and measured immunoglobulins differentiated by subclasses. The most prominent signs for groups with associated diabetic fetopathy are defined by significant alternation in IgA2 and IgG4 concentrations. All concentrations are defined in mg/mL units.

Acc. Number	Gene Name	Protein Name	G01, GDM	G02, GDM + DF	G03, T2DM	G04, T2DM + DF	G05, Control	*p*-Value †
P01857	IGHG1	Immunoglobulin IgG1	8.84 ± 1.22	7.81 ± 1.69	8.26 ± 2.09	7.65 ± 2.13	7.07 ± 1.99	1.032 × 10^−^^3^
P01859	IGHG2	Immunoglobulin IgG2	2.27 ± 0.41	2.43 ± 0.37	2.54 ± 0.31	2.37 ± 0.22	2.11 ± 0.14	0.774 × 10^−^^3^
P01860	IGHG3	Immunoglobulin IgG3	2.06 ± 0.49	2.12 ± 0.38	1.87 ± 0.24	2.05 ± 0.38	1.93 ± 0.65	0.591 × 10^−^^2^
P01861	IGHG4	Immunoglobulin IgG4	2.12 ± 0.42	1.85 ± 0.36	2.08 ± 0.49	1.81 ± 0.41	2.64 ± 0.98	1.065 × 10^−^^3^
P01871	IGHM	Immunoglobulin IgM	1.87 ± 0.31	1.67 ± 0.20	1.93 ± 0.26	1.71 ± 0.17	1.34 ± 0.12	4.912 × 10^−^^3^
P01876	IGHA1	Immunoglobulin IgA1	1.42 ± 0.26	1.98 ± 0.14	1.39 ± 0.19	1.55 ± 0.28	1.31 ± 0.23	2.817 × 10^−^^4^
P01877	IGHA2	Immunoglobulin IgA2	0.41 ± 0.08	0.52 ± 0.04	0.38 ± 0.07	0.46 ± 0.07	0.23 ± 0.09	1.296 × 10^−^^3^

† *p*−value was calculated by Kruskal–Wallis test at *p* < 0.05 significance; the indicated in Table *p*-value designates difference between all studied groups; statistical significance between the particular groups, if appropriate, is indicated in the text.

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
