# Peer review of "Association of Proteins Modulating Immune Response and Insulin Clearance during Gestation with Antenatal Complications in Patients with Gestational or Type 2 Diabetes Mellitus"

_cells, 2020, doi:10.3390/cells9041032_

Round 1

Reviewer 1 Report

This manuscript has been improved.

Author Response

Dera Reviewer and Lena Ilic,

Please, find the enclosed Letter with Replies to the Reviewer's commentaries. The letter contains point-by-point replies where it appropriate, and all corrections are provided in the revised version of the manuscript. Please, note, that we used "Tracking changes" mode while correcting the text, however, the most important and substantial changes were color-filled (yellow color) for attending and easier inspection.

On behalf of all co-authors,
Arthur T. Kopylov

Reviewer 2 Report

In the study the authors performed a proteomic and quantitative measurement of selected proteins from plasma samples collected from patients with maternal diabetes and from patients with uncomplicated pregnancy and demonstated that patients affected by maternal diabetes that gave birth to newborns with signs of fetopathy syndrome are characterized by espression of group-specific proteins.

I consider the examined topic relevant and interesting.

In comparison to the previously published literature, in the paper the authors try to identify one or more biomarkers that can be detected during fetal life and to predict poor fetal outcome. This may allow us to identify diabetic fetopathy in an eary gestational age with a simple blood sample in course of routine clinical analysis.

One limit of the study is the small cohort examined and this should be mentioned by the authors.

While the paper can be easily understood, the English language needs extensive revision.

Author Response

(The authors gave the same response as above.)

Reviewer 3 Report

In attachement the file with comments

Author Response

(The authors gave the same response as above.)

Reviewer 4 Report

This study aims at the discovery of biomarkers for early recognition of diabetic fetopathy (DF) as a complication in patients with diabetes mellitus during gestation which proposed title is “Association of proteins modulating immune response and insulin clearance during gestation with antenatal complications in patients with gestational or type 2 diabetes mellitus.” The authors did a proteomics evaluation in patients with gestational diabetes mellitus and type 2 diabetes mellitus.

General comments:

The manuscript was very hard to read, for example within methods the authors already have table 1 that should be in the results section and figure 4 (a result) in the discussion section.  In general, it is too extensive, not objective, and needs to be all revised.   Nevertheless, the authors show interesting results which should be objectively described.

  • The introduction is excessively long and not objective.
  • The authors address the very interesting hypothesis that insulin clearance is compromised in gestational diabetes. However, insulin clearance is not measured in the cohort. Unless the authors can present these results, which could really enrich the manuscript, the association of CEACAM and insulin clearance can only be mention in the discussion section as a hypothesis.
  • Table 2 is quite confusing as it appears to not address the proteins enrolled in diabetic fetopathy and they do not include CEACAM, CPR, etc. Again, it appears a collection of data without a clear rationale.
  • The discussion section is extensive, presents results and needs to be restructured.
  • The authors present typos through the manuscript even a sentence that is not finished (ex line 454: there were CDKN2A, CDKN2B, HHEX, ENPP1, PPARG, and plenty complement and 454 coagulation factors but almost.). Another example is CAECAM instead of CEACAM in the abstract. Indeed, the impression is that the present manuscript is a very initial draft.

Author Response

(The authors gave the same response as above.)

Round 2

Reviewer 3 Report

Comments are attached

Author Response

Dear Editor and Reviewer, please, find attached reply with point-by-point responses to the Reviewer's comments.

On behalf of all co-authors, 

Arthur T. Kopylov

Reviewer 4 Report

The manuscript improved substantially and only minor corrections have to be performed after the overall manuscript is redone.

Sincerely,

Author Response

Dear Editor and Reviewer, please, find attached reply with point-by-point responses to the Reviewer's comments.

On behalf of all co-authors, 

Arthur T. Kopylov

This manuscript is a resubmission of an earlier submission. The following is a list of the peer review reports and author responses from that submission.

Round 1

Reviewer 1 Report

This is an interesting topic but the text is difficult to understand. Results and methods have not been clearly presented. Conclusions and results are mixed. The abbreviations need better explanation (e.g. in the abstract). The article needs severe shortening and sharpening.

Introduction:
The effectiveness of treatment after early detection of DF needs to be explained. Why is early detection needed?

Results:
It is difficult to understand how the proteins are clustered. What is the basis for clustering the Proteins to the Group “positive regulation of macromolecule metabolic processes”? The expression “GO-clustering” has not been described.

Discussion:

The authors write: “Carnosine dipeptidase-1 is found mainly in muscle tissue”. This is certainly not true for mice or humans. Please correct or give the reference accordingly. Instead, carnosine, the Substrate for carnosine dipeptidase 1 is mainly found in muscle tissue.

Speculation and results are mixed, e.g. is there any scientific confirmation of upregulation of CNDP1 by increasing lactate Levels?

Reviewer 2 Report

The manuscript entitled “Association of proteins modulating immune response and insulin clearance during gestation with antenatal complications in patients with gestational or type 2 diabetes mellitus.” by Olga Papysheva1 and collaborators, aims to search for potential biomarkers for early prediction of diabetic fetopathy (DF). I will recommend some modifications before accepting it. Major comments: 1. The timing of the subjects’ blood collection was not found. It is a key issue. 2. I suggest that the authors list the 48 defined proteins of UPS-2. The author did not state the reason for choosing these 48 proteins. In addition, in results section, “proteomic surveying revealed 321 proteins shared between all study groups.” (line 27). What does it mean to characterize them? Aren't they just 48 proteins? 3. It is best to list statistics as a separate item in the Materials and Methods section. Not easy to find it for the analysis of the data of Tables 3 and 4. I did also not find the statistical methods in Table 1, including the comparative analysis of BMI, age, and OGTT test value. 4. I suggest the authors move Tables 3 and 4 to the results section. The Discussion section should focus on discussion. However, the authors write a lot about the results. The discussion section needs to be streamlined. 5. The title of Figure 3 is inappropriate. Also, references or sources of the signal transduction pathway were missed. 6. Recently, there have been some studies on blood proteomics in GDM by using iTRAQ and TMT-based isobaric labeling. The author can pay attention it. In Discussion section, regarding the mechanism, the author should not do too much speculative analysis. 7. The authors did not use other methods to validate the proteomics results. Why? Minor comments: 1. In Abstract section, the phrase “a number of” is inappropriate. (Page 1, line 26) 2. In Introduction section (Page 2, lines 84-89), although no existing methods provide sufficient and reliable signs of the emerging DF during pregnancy, are there any other studies on this issue? 3. There are no diagnostic criteria for type 2 diabetes in the “Inclusion criteria”. 4. Are the number of samples in Tables 3 and 4 the same as in Table 1? 5. Page 14, line 514, abbreviation “ROC” should give a full name. 6. Lines 387 and 396, the valence state of the ion should be superscript. 7. The manuscript needs to be carefully revised to eliminate the grammatical errors and sentence corrections.